# The Impact of Infrastructure Development on China–ASEAN Trade-Evidence from ASEAN

**Chen Shen**

College of Northeast Asian Studies, Jilin University, No. 2699 Qianjin Street, Changchun 130012, China; shenchen19@mails.jlu.edu.cn

**Abstract:** Abstract:From the formal dialogue between China and ASEAN in 1991 to the establishment of the China–ASEAN FTA in 2010, the economic and trade relations between China and ASEAN countries have developed rapidly. With the continuous development of economic and trade relations, the infrastructural level between China and ASEAN has far lagged behind the needs of exchanges and trade. How to promote the development of bilateral trade through infrastructure construction and interconnection has become a concern of governments of all countries. In the context of China's "Belt and Road" Initiative and the "Master Plan for ASEAN Connectivity", new ideas and opportunities are provided for ASEAN infrastructure construction and its interconnection with China. Based on panel data from 2004 to 2020, this paper analyzes the impact of ASEAN infrastructure on the China–ASEAN trade volume. The analysis finds that road, port, shipping and communication infrastructure positively impact the bilateral trade volume, among which ports exert the greatest impact on the bilateral trade volume and roads have the least impact. For a more in-depth study, the transportation infrastructures of land and island countries are compared and analyzed separately. The impact of road infrastructure on trade is significant for land countries, while the impact of port infrastructure on trade is more remarkable for island countries. Finally, measures and suggestions on how to promote the development of bilateral trade are proposed on the basis of the above analysis.

**Keywords:** interconnection; infrastructure; gravity equations; bilateral trade

## 1. Introduction

Over the past 30 years, since the establishment of a dialogue between China and ASEAN, economic and trade relations have grown rapidly. On 1 January 2010, the China–ASEAN Free Trade Area was officially founded as one of the fastest-growing regions in the contemporary world; in 1991, the bilateral trade volume between China and ASEAN was a mere USD 6.3 billion, while by 2020, the China–ASEAN overall trade volume had exceeded USD 684.6 billion. With economic and trade development, ASEAN's demand for infrastructure investment is rapidly increasing and intra-regional connectivity has become a key point in the completion of the new China–ASEAN economic landscape. The Master Plan on ASEAN Connectivity was unanimously adopted and signed by ASEAN leaders in 2010. In September 2016, ASEAN leaders agreed on the Master Plan on ASEAN Connectivity 2025 (MPAC 2025), a strategic guidance document built on the MPAC to further improve the region's connection and integration and re-emphasize the importance of connectivity. At the same time, China is also actively participating in China–ASEAN connectivity projects and supports ASEAN in implementing the Master Plan on ASEAN Connectivity as a new highlight in developing China–ASEAN economic cooperation.

In 2013, Chinese President Xi Jinping put forth major initiatives to jointly build "the Silk Road Economic Belt" and the "21st-Century Maritime Silk Road". Countries along the Belt and Road have rich natural resources, and the different resource endowments and their economies are highly complementary, which indicates excellent potential for economic and trade cooperation among them. Infrastructure interconnection is the priority area of the

Belt and Road and a bottleneck that restricts China–ASEAN cooperation. The development of economic and trade relations between the two sides needs to be supported by a more efficient and convenient infrastructure network. A joint statement on the dovetailing cooperation between the Belt and Road Initiative and the ASEAN Connectivity Master Plan 2025 was issued during the 22nd China–ASEAN Leaders' Meeting in Bangkok, Thailand, on 3 November 2019. The urgent issue in achieving interregional connectivity is the development of infrastructure.

There are great potential and practical conditions for China–ASEAN economic and trade relations to achieve a great leap. In the face of the increasingly complicated international situation and the sluggish recovery of the global economy and trade under the influence of COVID-19, China and ASEAN urgently need to promote the construction of connectivity, thus reducing trade costs and improving trade efficiency. Most ASEAN countries are lagging behind in infrastructure development. The lagging infrastructure limits the balanced economic development of the countries. China is currently the world's third-largest foreign investor and its construction capability in railways, highways, bridges, tunnels and other engineering fields has reached the world's most advanced level. China can and is willing to contribute to ASEAN infrastructural development and economic growth.

A literature review shows that the development of transport infrastructure through the Belt and Road Initiative can positively affect the trade between China and its partners. However, more research needs to be done on ASEAN. Based on the above background, the impact of ASEAN infrastructure development on China–ASEAN trade is studied in this article. The empirical analysis is conducted by adding infrastructure development indicators to the gravity model. The regression analysis concludes that infrastructure development related to airport quality and the share of Internet users significantly contributes to bilateral trade development. The establishment of a free-trade zone and land connectivity can effectively promote bilateral trade. Road construction in land-based countries has a more dramatic impact on China–ASEAN trade than that in island countries. The impact of port construction on trade is more pronounced in island countries than in land-based countries.

## 2. Literature Review

The study of the relationship between trade and infrastructure is not a new phenomenon. The significance of a country's infrastructure in promoting trade development cannot be ignored [1]. Some studies have found supporting evidence for a link between infrastructure and international trade growth using regression models [2,3]. More and better infrastructure can reduce trade-related transaction costs and facilitate international trade relations [4]. Infrastructure can significantly increase the total volume of international trade flows, with the most substantial impact on exports [5]. Improved infrastructure endowment and quality can reduce trade costs and increase international trade flows [6]. Brooks and Hummels (2009) [7] point out that infrastructure development plays a vital role in the expansion of trade in Asia. The quality and quantity of infrastructure in Central Asia positively impact trade [8].

Different types of infrastructure can have different impacts on trading partners [9]. Therefore, the trade effects of infrastructure also vary depending on the mode of transport. Since policymakers often need to decide which types of infrastructure should be improved, it is essential to further analyze the exact effects of improvements to specific infrastructure types. Several studies also confirm that improvement in transport infrastructure is a major factor in reducing the cost of international trade [3,10]. Limao and Venables (2001) [3] used the gravity equation to show that transport infrastructure increases the total trade volume and underdeveloped infrastructure in transit countries can impede trade flows. Wilson, Mann and Otsuki (2005) [11] estimated that port efficiency also has a positive trade impact on the total trade flows of manufactured goods. Blonigen and Wilson (2008) [12] also found that improvements in port efficiency increase the U.S. trade volume. Using a gravity model, Shepherd and Wilson (2008) [13] found that trade flows in Southeast Asia are particularly sensitive to transport infrastructure and information and communications technology.

Alderighi and Gaggero (2017) [14] demonstrated that an increase in direct flights also boosts export. Donaubauer, Meyer and Nunnenkamp (2016) [15] decomposed infrastructure into four components, namely, transportation infrastructure, information and communication technology (ICT) infrastructure, energy infrastructure and financial infrastructure. Among them, transportation infrastructure and financial infrastructure have the most significant trade effects. Bankole et al. (2015) [16] found a negative impact of telecommunication costs on trade through a gravity model.

The host country's infrastructure can also exert a significant impact on the trade volume of trading partners. The development of transport infrastructure through the Belt and Road Initiative can remarkably reduce transport time trade costs [17]. Regarding transportation infrastructure, Zhai (2018) [18] stated that B&R would improve transportation infrastructure, promoting trade, tourism and investment and cooperation with China. Chen, Z. (2021) [19] provided an in-depth assessment of transport infrastructure investment in the Belt and Road using a general equilibrium (CGE) model and concluded that transportation infrastructure investments effectively reduce trade costs. Wang (2018) [20] declared that the expansion of infrastructure and improved logistics performance in 48 countries along the Belt and Road has had a positive impact on China's exports. Li et al. (2016) [21] found that intercontinental railroads positively affect the trade between China and its partners.

In summary, currently, scholars have analyzed the impact of infrastructure on trade from theoretical and empirical perspectives, respectively. In recent years, ASEAN has made infrastructure construction and connectivity a key development area; meanwhile, ASEAN is a priority region for China's Belt and Road Initiative. By classifying ASEAN infrastructure into different types and conducting quantitative analysis, it will be more comprehensive and intuitive to study its role in bilateral trade. It is of great realistic significance for promoting the compelling connection between MPAC 2025 and the Belt and Road Initiative.

## 3. Trade Analysis

Since 2010, ASEAN trade volume grew from USD 201.443 billion to USD 291.23 billion by 2020. The trade volume between ASEAN and the United States increased from USD 181.241 billion in 2010 to USD 301.097 billion in 2020. The trade volume between ASEAN and Japan decreased from USD 218.927 billion in 2010 to USD 194.89 billion in 2020. Since the establishment of the China–ASEAN Free Trade Area in 2010, China has remained the top trading partner of ASEAN for 12 consecutive years. Moreover, bilateral trade volume grew from USD 292.776 billion in 2010 to USD 684.6 billion in 2020. The volume of trade grew by almost USD 391.9 billion in ten years. However, the growth rate of bilateral trade has been gradually declining, with negative growth for the first time in 2015. With the implementation of the ASEAN Connectivity Master Plan and the promotion of China's Belt and Road initiative, bilateral trade has gradually resumed positive growth. ASEAN has become one of China's major trading partners. China is a preferred market for ASEAN.

Despite the negative impact of COVID-19, China–ASEAN trade in goods still achieved relatively rapid growth against the trend. Bilateral trade reached USD 684.6 billion by 2020. In 2020, bilateral trade volume reached USD 684.6 billion, with a year-on-year increase of 6.7%. China's exports to ASEAN amounted to USD 383.72 billion and imports from ASEAN reached USD 300.88 billion. The bilateral trade volume between China and ASEAN accounted for 14.7% of China's total foreign trade and 19.4% of ASEAN's total trade volume in 2020. Moreover, ASEAN has surpassed the European Union to become China's largest trade partner in goods for the first time in history (Table 1).

**Table 1.** China–ASEAN Trade Statistics 2010–2020 (Unit: USD billion).

| Year | Trade Volume | Growth Rate (%) | Exports | Growth Rate (%) | Imports | Growth Rate (%) |
|------|--------------|-----------------|---------|-----------------|---------|-----------------|
| 2010 | 292.776 | 37.5 | 138.207 | 30.1 | 154.569 | 44.8 |
| 2011 | 362.854 | 23.9 | 170.083 | 23.1 | 192.771 | 24.6 |
| 2012 | 400.09 | 10.2 | 204.27 | 20.1 | 195.82 | 1.5 |
| 2013 | 443.61 | 10.9 | 244.09 | 19.5 | 199.54 | 1.9 |
| 2014 | 480.39 | 8.3 | 272.07 | 11.5 | 208.32 | 4.4 |
| 2015 | 472.16 | −1.7 | 277.49 | 2 | 194.68 | −6.6 |
| 2016 | 452.2 | −4.1 | 255.99 | −7.7 | 196.22 | −0.9 |
| 2017 | 514.82 | 13.8 | 279.12 | 9 | 235.7 | 20.1 |
| 2018 | 587.87 | 14.1 | 319.24 | 14.2 | 268.63 | 13.8 |
| 2019 | 641.46 | 9.2 | 359.42 | 12.7 | 282.04 | 5 |
| 2020 | 684.6 | 6.7 | 383.72 | 6.7 | 300.88 | 6.6 |

Source: According to China Customs statistics.

Except for a temporary decline in China's trade with Indonesia due to COVID-19, China's trade with other ASEAN countries has generally been on the rise. However, there is a massive gap in the trade volume between China and ASEAN member countries. Among the 10 ASEAN countries, Vietnam has been China's largest trading partner for six consecutive years, with bilateral trade volume reaching USD 192.289 billion. The combined trade volume of Vietnam and Malaysia with China accounted for 47.2% of the total China–ASEAN trade volume.

Since the 21st century, China's trade with other less-developed countries such as Cambodia, Laos and Myanmar has been growing steadily, but the total volume is still low. According to China's Ministry of Commerce, the bilateral trade volume between China and Myanmar was USD 18.7 billion in 2019 and the bilateral trade volume between China and Laos was only USD 3.92 billion in 2019 (Figure 1).

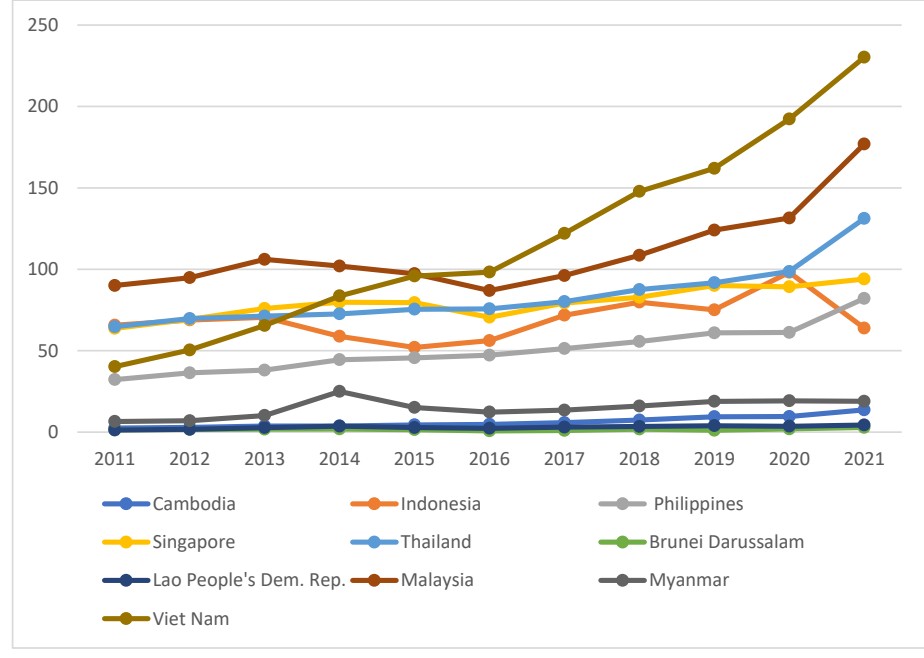

**Figure 1.** Trade volume between China and ASEAN countries 2011–2021 (Unit: USD billion).

From the above analysis, it is clear that the trade volume between China and ASEAN countries has been growing rapidly. In the face of the current impact of COVID-19, sluggish global economic growth, weak demand and rising trade protectionism, China can actively participate in China–ASEAN connectivity projects and support ASEAN in implementing the Master Plan on ASEAN Connectivity, thus reducing trade costs and improving trade efficiency.

## 4. Research Methodology

### 4.1. Model Specification and Variable Description

To understand the impact of different types of infrastructure on China's trade with ASEAN, the gravity model is adopted. Tinbergen (1962) [22] introduced the first gravity equation model to study international trade issues. Subsequently, the gravity equation model became one of the most widely used tools by researchers to analyze international trade flows and their determinants. The gravity equation model states that trade flows between two countries depend on the quality of their economies and the distance between them. After a period of development, the gravitational model has also been continuously improved. The basic form of the gravitational model is:

$$T_{IJ} = K \left( \frac{GDP_i^{\alpha_1} GDP_j^{\alpha_2}}{Dist_{ij}^{\beta}} \right) e_{ij} \tag{1}$$

$T_{IJ}$ is the total trade volume. Subscripts i and j signify two different countries or regions, respectively. $GDP_i$ represents the gross domestic product of country i. $GDP_j$ represents the gross domestic product of country j. $Dist_{ij}$ denotes the distance between country i and country j. $\alpha_1$, $\alpha_2$ and $\beta$ are the parameters. $e_{ij}$ is the disturbance term with expectation 1.

In economic research, regression analysis using only cross-sectional data would leave out time-varying features, while analysis using only time series would leave out the linkages and differences between cross-sections. Panel data can analyze a large amount of data. They can enhance the model's degree of freedom and reduce the multicollinearity among explanatory variables. Therefore, the panel data can better analyze the trade between China and 10 ASEAN countries from 2004 to 2020.

Infrastructure measurement indicators are further refined based on previous studies; that is, in the empirical analysis, the infrastructure is first divided into transportation infrastructure and communication infrastructure and then the transportation infrastructure is subdivided into roads, airlines and ports. The presence or absence of road connectivity with China and the establishment of a China–ASEAN FTA are added to the gravity model as dummy variables. Combining theoretical and empirical studies, the model is proposed as follows:

In

$$T_{ijt} = \alpha_0 + \alpha_1 \ln GDP_{it} + \alpha_2 \ln GDP_{jt} + \beta_1 \ln Dist_i + \beta_2 \ln Road_{it} + \beta_3 \ln Air_{it} + \beta_4 \ln Port_{it} + \beta_5 \ln Int_{it} + \beta_6 FTA_{it} + \beta_7 Con_i + \mu \tag{2}$$

$T_{ijt}$ denotes the trade volume of country i with country j (China). $GDP_{it}$ denotes the GDP of country i. $GDP_j$ denotes the GDP of China. $Dist_i$ denotes the straight-line distance between the capital of country i and Beijing (Capital of China). $Road_{it}$ indicates the total mileage of national roads in ASEAN countries. $Air_{it}$ represents the number of international airports (in count). $Port_{it}$ represents the number of international ports (in count). $Int_{it}$ indicates individuals using the Internet (% of population). The establishment of the China–ASEAN Free Trade Area in 2010 is expressed by $FTA_{it}$. The establishing $FTA_{it}$ value is 0, otherwise, its value is 1. $Con_i$ says ASEAN countries are interconnected with roads in China. The value is 1 if there is a road connection with China, otherwise, the value is 0.

### 4.2. Data and Descriptive Statistics

Data are used to estimate our model and all data are collected from online sources, including websites of national agencies and organizations concerned. The primary data sources include the World Bank, UN Comtrade Database and AEANStatsDataPortal. The distance between China and ASEAN countries refers to the straight-line distance between the capitals of each country. The data source website is http://www.mapcrow.info/ (accessed on 10 May 2022). Details of the specific data sources are as follows. This table does not include other dummy variables (Table 2).

**Table 2.** List of Variable and Data Sources.

| Variable Name | Abbreviation | Data Source |
|---|---|---|
| Trade value | $T_{ijt}$ | UN Comtrade Database |
| Distance between capital cities of trading partners | $Dist_i$ | http://www.mapcrow.info/ |
| Gross domestic product (country i and country j) | $GDP_i$, $GDP_j$ | World Bank Open Data |
| Length of roads | $Road_{it}$ | ASEANStatsDataPortal |
| Number of international ports | $Port_{it}$ | ASEANStatsDataPortal |
| Number of international airports | $Air_{it}$ | ASEANStatsDataPortal |
| Individuals using the Internet | $Int_{it}$ | World Bank Open Data |

Table 3 shows a summary of the descriptive statistics for all variables of the model, including dependent and independent variables. The table provides a brief overview of the different statistical outputs of the variables. The results are based on 10 variables, nine of which are independent variables and trade is the dependent variable.

**Table 3.** Descriptive statistics.

| Variable | Obs | Mean | SD | Minimum | Maximum |
|---|---|---|---|---|---|
| $T_{ijt}$ | 170 | 37,800 | 21,603,870 | 11.4 | 19,200 |
| $GDP_{it}$ | 170 | 21,600 | 24,400 | 237 | 112,000 |
| $GDP_{jt}$ | 170 | 823,365 | 425,864 | 196,168 | 1,476,890 |
| $Dist_i$ | 170 | 3551.468 | 867.3437 | 2324.8 | 5217.57 |
| $Road_{it}$ | 170 | 162,425.7 | 179,320.9 | 2512.163 | 702,576.5 |
| $Port_{it}$ | 170 | 56.47059 | 84.62421 | 0 | 396 |
| $Air_{it}$ | 170 | 7.047059 | 7.8671 | 1 | 34 |
| $Int_{it}$ | 170 | 35.69512 | 28.05494 | 0.0243374 | 95 |
| $FTA_{it}$ | 170 | 0.6470588 | | 0 | 1 |
| $Con_i$ | 170 | 0.5 | | 0 | 1 |

The independent variables are further divided into two parts: seven main variables and two dummy variables. Based on the panel data for 10 ASEAN countries from 2004 to 2020, the sample consists of 170 observations.

The average value of China's trade with ASEAN trading partners is USD 37,800 million, representing the trade flows in the model. The table shows that China's GDP is much higher than its trading partners. One of the main reasons may be China's larger population and higher development level, leading to an increase in all other economic factors. The average distance between China and the capitals of its trading partners is 3551.468 km, with a minimum distance of 2324.8 km and a maximum distance of 5217.57 km. Some ASEAN countries, such as Vietnam and Laos, are very close to China and are connected by land. The statistical table also summarizes the variables of the transport and communication infrastructure. Regarding transportation infrastructure, the average length of roads in

the sample of ASEAN countries is 162,425.7 km, with a minimum value of 2512.163 km and a maximum of 5217.57 km. Similarly, the study's results also show that, in terms of transportation infrastructure (road, maritime, air), road infrastructure has the highest average coverage, while waterways have the lowest average coverage. Meanwhile, as Laos is a landlocked country, the number of international ports is zero. Therefore, for landlocked countries like Laos, transportation infrastructure needs to be strengthened to compensate for the lack of ports in these landlocked countries. Some ASEAN countries gained access to Internet services in the first decade of the 21st century and reached 95% per capita. However, Internet use still needs to improve in rural areas of most ASEAN countries.

*4.3. Empirical Results*

Panel data are now being used increasingly in economic research. Panel data are better suited to study the dynamics of change [23]. In regression analysis, putting all the data together in an OLS regression is called mixed regression. The model form is as follows: $y_{it} = \alpha + x'_{it}\beta + u_{it}$. The underlying assumption of mixed regression is that there are no individual effects. Therefore, it is difficult to find real-life situations where the effect of the explanatory variables on the explained variables is independent of the cross-sectional individual. The fixed effects model is as follows: $y_{it} = x'_{it}\beta + \alpha_i + u_{it}$.

The model assumptions indicate that individual characteristics are correlated with explanatory variables. $\alpha_i$ denotes the respective different constant terms for each individual in the regression model. The fixed effects model assumes different intercepts between individuals in the cross-section and the same slope coefficient. The random effects model is as follows: $y_{it} = x'_{it}\beta + \alpha + \varepsilon_i + u_{it}$. The random effects model assumes that $\varepsilon_i$, $u_{it}$ and $x'_{it}$ are uncorrelated. $\varepsilon_i$ is a random element of cross-sectional individual influence, for each cross-section individual is not the same. The fixed effects model considers the error term and the explanatory variables to be correlated. The random effects model assumes that there is no correlation between the error term and the explanatory variables. Since the ASEAN countries studied in this paper have different characteristics, mixed regression models are excluded. When dealing with panel data, it is necessary first to conduct a test to select a model suitable for the panel data in this paper.

In selecting fixed and random effects models, the Hausman (1978) [24] test is generally used for determination. The principle of the Hausman test is to determine the choice of a data model by testing whether the model error term is orthogonal to the explanatory variables. Suppose the error term of the test model is orthogonal to the explanatory variables, i.e., $E\{x'_{it}\varepsilon_i\} = 0$; the model should be set as a random effects model, otherwise set as a fixed effects model. According to Hausman, the test *p* value is 0.001. The fixed effects model is the best-estimated model that can be used. Table 4 presents the gravity model with the introduction of infrastructure variables.

Model 1 is the same as expected, with a positive GDP sign for China and ASEAN countries. The economic aggregates of each country are positively correlated with the bilateral trade volume. For every 1% increase in the GDP of ASEAN countries, China's bilateral trade with ASEAN will increase by 1.12%. It shows that the economic development of ASEAN countries will significantly contribute to the development of bilateral trade.

Most of the ASEAN countries are coastal countries and shipping has the characteristics of low cost and large transport volume. The impact of port infrastructure development on bilateral trade is positive and significant. For every 10% increase in port infrastructure development in ASEAN countries, bilateral trade will increase by 7.9%. Although air cargo volume is smaller, it is more efficient. Therefore, the construction of aviation infrastructure also has a noticeable impact on imports and exports. Strengthening aviation infrastructure can promote the development of bilateral trade between China and ASEAN.

**Table 4.** Introduction of gravity model with infrastructure variables.

| | Total Samples | | Land Countries | | Island Countries | |
|---|---|---|---|---|---|---|
| | Model 1 | Model 2 | Model 3 | Model 4 | Model 5 | Model 6 |
| $\ln GDP_{it}$ | 1.121 *** | 0.274 ** | 0.319 *** | 0.425 ** | 0.635 *** | 0.315 ** |
| | (0.277) | (0.153) | (0.032) | (0.154) | (0.063) | (0.187) |
| $\ln GDP_{jt}$ | 0.314 *** | 0.391 *** | 1.013 *** | 1.153 * | 0.736 *** | 0.163 *** |
| | (0.166) | (0.093) | (0.021) | (0.153) | (0.054) | (0.073) |
| $\ln Dist_i$ | −0.977 *** | −0.667 ** | −0.463 ** | −0.531 ** | −0.673 *** | −0.631 ** |
| | (0.164) | (0.326) | (0.421) | (0.127) | (0.361) | (0.563) |
| $\ln Road_{it}$ | | 0.154 ** | 0.338 *** | | 0.151 ** | |
| | | (0.715) | (0.245) | | (0.243) | |
| $LnAir_{it}$ | | 0.132 ** | 0.345 ** | 0.251 *** | 0.264 *** | 0.0215 * |
| | | (0.258) | (0.314) | (0.157) | (0.251) | (0.146) |
| $\ln Port_{it}$ | | 0.798 *** | | 0.078 ** | | 0.875 *** |
| | | (0.127) | | (0.015) | | (0.068) |
| $\ln Int_{it}$ | | 0.354 * | | | | |
| | | (0.075) | | | | |
| $FTA_{it}$ | | 0.274 *** | | | | |
| | | (0.217) | | | | |
| $Con_i$ | | 0.164 *** | | | | |
| | | (0.261) | | | | |
| cons | −6.269 | −0.3153 | −4.23 * | −7.43 * | −5.372 | −1.573 |
| | (0.273) | (0.573) | (0.052) | (0.052) | (0.748) | (0.683) |
| wald chi2 | 1282.50 | 1425.77 | 1688.60 | 2179.01 | 442.07 | 395.73 |
| obs | 170 | 170 | 85 | 85 | 85 | 85 |

Note: * is significant at the 10% level, ** is significant at the 5% level, *** is significant at the 1% level and the numbers in parentheses are the standard deviations of the regression coefficients of the variables.

Model 2 shows that the development of communication infrastructure, such as the Internet, has a tremendous impact on bilateral trade flows. The development of the Internet has shortened the distance between producers and consumers worldwide, reduced transaction costs for businesses and increased the productivity of workers. In the era of e-commerce, communication facilities such as the Internet greatly reduce trade costs and facilitate business communication. Infrastructure development, such as communications, has considerably facilitated bilateral trade. In Model 2, two dummy variables are added to whether the China–ASEAN FTA is established and whether there is a road connection with China on land. The establishment of the FTA has had an enormous impact on trade. Since the establishment of the China–ASEAN Free Trade Area in 2010, China has remained the top trading partner of ASEAN for 12 consecutive years, and bilateral trade volume grew from USD 292.776 billion in 2010 to USD 684.6 billion in 2020. It shows that strengthening policy docking and removing trade barriers can promote bilateral trade development. The interconnection of land roads can effectively eliminate the harmful effects of borders and distances. For instance, trade with Vietnam is much higher due to road connection than estimated using only the other variables. The scale of trade between Vietnam and China is increasing by more than 10% every year.

Model 2 shows that the road construction of ASEAN countries has little impact on bilateral trade. By sorting out the geography of ASEAN countries, it can be found that Cambodia, Thailand, Laos, Vietnam and Myanmar are land countries, while Malaysia, Singapore, Indonesia, Brunei and the Philippines are island countries. While ASEAN land countries can trade with China by land transport, ASEAN island countries can only trade goods via sea and air transport. Thus, the development of different transportation infrastructures may play a different role in the trade of ASEAN land and maritime countries with China. This situation will be further analyzed below.

Table 4 analyzes the impact of transport infrastructure on trade in ASEAN land and maritime countries, respectively. Comparing Model 3 with Model 5 shows that road construction in land countries significantly impacts China–ASEAN trade more significantly

than in maritime countries. This may be due to the fact that the construction of roads in island countries can only provide an excellent boost to the flow of domestic goods. Roads are not interconnected with China, so the impact on trade with China is not noticeable. Since all ASEAN land countries are connected to China by road, trade transport can be realized by land transport. Therefore, the construction of roads in land countries has a more substantial impact on bilateral trade than in island countries.

From a comparison of Model 4 and Model 6, it is clear that ASEAN land countries can trade with China through land transport, so the construction of ports does not have an effect on their trade with China as significant as in the island countries. ASEAN island countries trade with China mainly by sea, so port throughput significantly impacts bilateral trade.

For ASEAN land countries, road infrastructure impacts trade more than air and ports. Every 1% increase in road infrastructure will increase bilateral trade by 0.33%. For every 1% increase in port infrastructure, bilateral trade between China and ASEAN will increase by 0.07%. Therefore, ASEAN landlocked countries should prioritize road infrastructure.

For island countries, ports are compared with air infrastructure. At a significant level of 1%, every 1% increase in port infrastructure will increase bilateral trade by 0.87%. The impact of aviation infrastructure on trade is not as significant as that of ports. At a significant level of 10%, a 1% increase in aviation infrastructure is associated with a 0.02% increase in bilateral trade between China and ASEAN. Therefore, to promote the better development of bilateral trade between China and ASEAN, ASEAN island countries should prioritize the development of port infrastructure. For areas less accessible by sea, priority can be given to building aviation infrastructure.

## 5. Conclusions and Policy Implication

### 5.1. Conclusions

From the ASEAN infrastructure perspective, this paper attempts to analyze the impact of ASEAN infrastructure development on its bilateral trade with China. The empirical analysis is conducted by adding infrastructure development indicators to the gravity model. The regression analysis concludes that infrastructure development related to air cargo volume, port throughput and the share of Internet users significantly contributes to bilateral trade development. When two dummy variables for road connectivity and being a free trade zone establishment are added, the analysis reveals that establishing a free trade zone and land connectivity can effectively promote bilateral trade.

Due to different geographical environments, ASEAN countries can be divided into land and island countries. To conduct a more in-depth study, the transportation infrastructure of ASEAN land and island countries are compared and analyzed, respectively. Road construction in land-based countries has a more dramatic impact on China–ASEAN trade than that in island countries. The effect of port construction on trade is more pronounced in island countries than in land-based countries. A comparison of various infrastructures in land-based countries shows that roads and ports have an equally significant impact on trade. Therefore, in terms of infrastructure development, ASEAN landlocked countries should give priority to the smooth flow of roads and improving the missing sections at the docking points. The development of railway freight between China and Laos, the construction of the Sino-Thai railway and the construction of railways and expressways between China and Vietnam will promote trade between China and neighboring countries.

Since port construction has a more serious impact on trade in island countries than in land-based countries, priority should be given to developing port infrastructure in island countries. Aviation infrastructure greatly impacts the trade of both ASEAN land and island countries, so the development of aviation infrastructure should be coordinated with the development of roads in the land countries. For island countries, there should be coordinated development of port and aviation construction, with coastal cities giving priority to the development of port facilities. The establishment of the China–ASEAN port city cooperation network will boost trade in ASEAN coastal countries. Instead, mainland cities should prioritize the construction of aviation facilities. Laos is landlocked and cannot

trade directly by sea, so road access has a tremendous impact on its trade. The construction of roads and railroads should be given priority.

Through comparative analysis, the result is that ASEAN countries should carry out corresponding constructions according to different situations. In this way, the barriers to bilateral trade between China and ASEAN can be reduced through infrastructure development so that trade potential can be well exploited.

### 5.2. Policy Implication

From the above analysis, it is clear that ASEAN infrastructure development plays a vital role in promoting bilateral trade between China and ASEAN. As the largest developing and most populous country in the world, China has a huge demand for raw materials and agricultural products, while most ASEAN countries have relatively poor infrastructure. Therefore, there is an urgent need for infrastructure development to promote economic development and improve national life. However, ASEAN infrastructure development and connectivity with China face many difficulties. China should combine the construction of the Belt and Road with the development plans of ASEAN countries to promote interconnection jointly.

China and ASEAN countries are connected by land and sea, with a land border of more than 4000 km. China has strong cultural ties with ASEAN countries. The interconnection between China and ASEAN is a systematic project, especially the infrastructure interconnection, involving a wide range of domestic and foreign stakeholders. By sorting out the geography of ASEAN countries, it can be found that Cambodia, Thailand, Laos, Vietnam and Myanmar are land countries. The government can use the railways to connect China, Malaysia, Laos and Thailand for the smooth flow of goods. It can be found that Malaysia, Singapore, Indonesia, Brunei and the Philippines are island countries. For island countries, China can support the construction of ports in ASEAN countries and establish a China–ASEAN port city cooperation network to achieve maritime connectivity. While promoting land-based connectivity, the development of maritime connectivity should be accelerated. To achieve efficient connectivity, China should increase its investment in less-developed ASEAN countries and support ASEAN to narrow down the internal development gap. It is also necessary to make more reasonable use of the China–ASEAN Maritime Cooperation Fund, Silk Road Fund and China–ASEAN Investment Cooperation Fund for ASEAN infrastructure development.

Governments should actively guide enterprises to participate in infrastructure development. Businesses and people are the biggest beneficiaries of connectivity; so it is also one of the essential forces driving interconnection. In particular, the participation of private enterprises can also eliminate the concerns among governments. Governments can provide companies with policy and tax support to encourage them to participate in the interconnection master plan. At the same time, strong and creditable enterprises of each country are encouraged to actively participate in the construction of ASEAN infrastructure. Enterprises should also actively play the allocation of resources to establish a good corporate image. For countries that lack capital, Chinese companies can carry out various forms of cooperation. Joint ventures can be selected according to the actual situation in the relevant countries and projects. The local government can pay the project company through resources or direct payments, etc.

ASEAN is an economy with more than 600 million people and a total area of over 4 million square kilometers. It is difficult for China and ASEAN alone to promote bilateral infrastructure development and connectivity effectively. Funds from international organizations can be actively used to help build infrastructure in the less-developed regions of ASEAN. China and ASEAN countries should reasonably use advanced technology and funds from Japan, the United States and other Western countries.

**Funding:** This research received no external funding.

**Institutional Review Board Statement:** Not applicable.

**Informed Consent Statement:** Not applicable.

**Data Availability Statement:** Data sharing is not applicable to this article.

**Conflicts of Interest:** The author declares no conflict of interest.

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
