# Peer review of "The Impact of Infrastructure Development on China–ASEAN Trade-Evidence from ASEAN"

_sustainability, doi:10.3390/su15043277_

Round 1

Reviewer 1 Report

see attached doc

Reviewer 2 Report

My main concerns are twofold. First, the paper needs to specify its goals, methods, and findings in the introduction, including the contribution made to the literature. Has this sort of empirical study ever been undertaken between China and ASEAN? If not, has it been done between China and other Belt and Road partners? 

Second, the description of the study that was undertaken and what it tells us that is new and intriguing needs to be bumped up significantly. We can surmise that building ports between close neighbours would stimulate trade but we don't really have a good decomposition of the data among the individual countries. Which countries would benefit more from which infrastructure development? If you can answer this question, that would be a really interesting paper.
